# The Impact of Resident Adipose Tissue Macrophages on Adipocyte Homeostasis and Dedifferentiation

**DOI:** 10.3390/ijms252313019

**Published:** 2024-12-04

**Authors:** Julia Neugebauer, Nora Raulien, Lilli Arndt, Dagmar Akkermann, Constance Hobusch, Andreas Lindhorst, Janine Fröba, Martin Gericke

**Affiliations:** 1Institute of Anatomy, Leipzig University, 04103 Leipzig, Germany; 2Paul-Flechsig-Institute, Leipzig University, 04103 Leipzig, Germany

**Keywords:** adipose tissue, dedifferentiation, DFAT, macrophages, inflammation, obesity

## Abstract

Obesity is concurrent with immunological dysregulation, resulting in chronic low-grade inflammation and cellular dysfunction. In pancreatic islets, this loss of function has been correlated with mature β-cells dedifferentiating into a precursor-like state through constant exposure to inflammatory stressors. As mature adipocytes likewise have the capability to dedifferentiate in vitro and in vivo, we wanted to analyze this cellular change in relation to adipose tissue (AT) inflammation and adipose tissue macrophage (ATM) activity. Using our organotypic AT explant culture method combined with a double-reporter mouse model for labeling ATMs and mature adipocytes, we were able to visualize and quantify dedifferentiated fat (DFAT) cells in AT explants. Preliminary testing showed increased dedifferentiation after tamoxifen (TAM) stimulation, making TAM-dependent lineage-tracing models unsuitable for quantification of naturally occurring DFAT cells. The regulatory role of ATMs in adipocyte dedifferentiation was shown through macrophage depletion using Plexxicon 5622 or clodronate liposomes, which significantly increased DFAT cell levels. Subsequent bulk RNA sequencing of macrophage-depleted explants revealed enrichment of the tumor necrosis factor α (TNFα) signaling pathway as well as downregulation of associated genes. Direct stimulation with TNFα decreased adipocyte dedifferentiation, while application of a TNFα-neutralizing antibody did not significantly alter DFAT cell levels. Our findings suggest a regulatory role of resident ATMs in maintaining the mature adipocyte phenotype and preventing excessive adipocyte dedifferentiation. The specific regulatory pathways as well as the impact that DFAT cells might have on ATMs, and vice versa, are subject to further investigation.

## 1. Introduction

During the past 50 years, obesity rates have continuously increased on a global scale, defining obesity as one of the most relevant issues in modern medicine [1]. Obesity as a multisystemic disease is associated with adipocyte hypertrophy and impaired adipogenesis, resulting in continuous activation of stress-related signaling pathways and subsequently impaired tissue function, which in turn leads to diseases such as cardiovascular dysfunction, immune disorders, insulin resistance, and type II diabetes mellitus [2,3]. In adipose tissue (AT), this immunological dysregulation presents as chronic low-grade inflammation mainly mediated through adipose tissue macrophages (ATMs) found in cell clusters (called crown-like structures (CLS)) surrounding dead adipocytes [4,5,6]. Nevertheless, it is vital to recognize ATMs as a heterogeneous cell population performing various, partly opposing functions, such as promoting inflammation as well as adipogenesis and AT homeostasis [7,8]. While not reflecting the full extent of macrophage subtypes and functions, ATMs are commonly categorized into the classically activated, pro-inflammatory M1 and the alternatively activated, anti-inflammatory M2 phenotype [9,10]. In obesity, the continuous inflammatory state is maintained by increased M1 polarization of resident tissue macrophages or through the recruitment of monocytes with subsequent differentiation and polarization towards an M1 phenotype [11,12].

Newer findings suggest that the continuous prevalence of oxidative stress, hypoxia, and inflammation results in impaired cell function not only through cell death but through reduced expression of mature cell markers and, thus, cells dedifferentiate into a precursor-like state, as observed in pancreatic β-cells [13,14]. In AT, dedifferentiation of mature adipocytes into fibroblast-like cells newly expressing stem cell genes and surface markers has been observed in experiments in vitro as well as in vivo [15,16,17]. In vitro adipocyte dedifferentiation is typically induced through a culture model established by Sugihara et al. titled “ceiling culture”. The buoyancy, adhesion, and confluence of isolated mature adipocytes in a media-filled culture flask results in a dedifferentiated fat (DFAT) cell culture after 7 to 10 days of incubation [18]. Recently, adipocyte de- and re-differentiation have also been observed in various physiological processes such as reconstruction of adipose tissue in murine mammary glands during and after lactation, wound healing, and the hair cycle in dermal AT [19,20]. While lipolysis, lipogenesis, and leptin secretion can still be stimulated to some extent in DFAT cells in vitro, they downregulate mature adipocyte genes and gain the ability to secrete a plethora of cytokines dependent on their dedifferentiation stage and initial AT depot [21,22,23,24]. With DFAT cells expressing multilineage re-differentiation potential, making them an interesting option for regenerative medicine as well as considering the potential role of DFAT cells in physiological AT adaptability, it is crucial to gain a better understanding of the underlying regulatory mechanism promoting adipocyte dedifferentiation [25,26,27,28]. Adipocyte dedifferentiation is an adaptive process, for which mechanical stress, adipose tissue remodeling, and tumorigenesis have been identified as influencing factors [29,30,31]. Additionally, transforming growth factor β (TGF-β), Wnt/β-catenin, and immunological signaling (found in inflammatory zone 1 (FIZZ1), tumor necrosis factor α (TNFα), and oncostatin M (OSM)), as well as insulin deficiency, have been suggested to induce adipocyte dedifferentiation [32,33,34,35,36,37,38,39,40]. With these signaling pathways being related to M1 and M2 ATM activity and known as regulatory pathways in AT, we hypothesized that resident ATMs are capable of affecting adipocyte dedifferentiation [7,35,39].

So far, insights regarding underlying regulatory mechanisms of adipocyte dedifferentiation have primarily been investigated using variants of the ceiling culture method [41,42]. While this enables the generation of a relatively pure DFAT cell population from isolated mature adipocytes, it does not account for the complex physiological environment present in AT, which affects the dedifferentiation process. Using our AT explant culture model, we were able to observe the emergence of DFAT cells from mature adipocytes integrated into the surrounding physiological stromal tissue structure [43]. Consequently, the model allowed the examination of correlations between resident ATM activity and DFAT cell occurrence.

In this work, we analyzed the effect of resident ATMs on adipocyte dedifferentiation in an AT explant culture model. Initially, we compared different mouse models for adipocyte visualization and found that tamoxifen (TAM) stimulation increased the number of DFAT cells, making TAM-based lineage tracing models unsuitable for the quantification of adipocyte dedifferentiation, at least in vitro. Macrophage depletion through two different approaches led to increased shares of DFAT cells in both setups. RNA sequencing of the AT identified the TNFα signaling pathway as a regulatory mechanism, which was further verified through TNFα stimulation. Our findings postulate a regulatory role for ATMs in adipocyte dedifferentiation, with macrophages inhibiting the excessive formation of DFAT cells and thereby mediating adipose tissue adaptability.

## 2. Results

### 2.1. Number of Dedifferentiated Adipocytes Increased over Time in AT Explant Culture

Previous experiments by our group with the established organotypic AT explant culture model coincided with the appearance of spindle- or stellate-shaped cells with various vacuoles and cytoplasmatic projections, morphologically similar to dedifferentiated adipocytes [43]. To verify this observation, we cultivated epididymal AT explants of lean, chow-fed homozygous AdipoqCreER^T2^ × ROSA26-CAG-tdTo × CSF1R-EGFP (MacFat) mice and analyzed living explants through microscopy as well as flow cytometry for quantification of tdTomato (tdTo)-expressing stroma cells on days 0, 7, and 10 of cultivation. The MacFat mouse model expresses the green fluorescent protein (GFP) continuously in resident AT macrophages, while tdTo is inducible through a Cre/loxP system controlled by the adiponectin promotor, which can be activated through TAM. However, the homozygous genotype of the MacFat mice used for these experiments previously showed a Cre leakage over 60%, allowing adipocyte visualization without prior induction and adverse effects of TAM, as outlined in detail below [44].

After 7 days of cultivation, a notable increase of stellate-shaped tdTo^+^ cells with several small vacuoles and partially interconnected cytoplasmatic projections was observed via microscopic imaging (arrow in Figure 1A). After 10 days, this increase was even more distinct. The subsequent flow cytometry analysis validated the microscopy results (Figure 1B). While the overall numbers of viable cells (4′,6-diamidino-2-phenylindole (DAPI)-negative proportion of measured single cells ≙ stromal vascular fraction (SVF); gating strategy in Appendix A) and ATMs (GFP^+^; tdTo^−^ proportion from SVF) remained constant (Figure 1C,D), the amount of red-fluorescent, fibroblast-like cells (GFP^−^; tdTo^+^ proportion from SVF) increased significantly, with the proportion nearly doubling by day 7 and tripling by day 10, respectively (Figure 1E). Simultaneously, the prevalent macrophage phenotype dramatically switched from the anti-inflammatory M2 (CD11c^−^; CD301^+^ proportion from ATMs) into a pro-inflammatory M1 (CD11c^+^; CD301^−^ proportion from ATMs) type during the 10-day cultivation period (Figure 1F). While on day 0 the ATM population consisted of almost exclusively M2 macrophages, the cell shares on day 7 had shifted in favor of the M1 macrophages (Figure 1G,H).

### 2.2. TAM Affected Adipocyte Dedifferentiation In Vitro, Making the Heterozygous MacFat Mouse Model Unsuitable for DFAT Cell Quantification in Explant Culture

The homozygous MacFat mouse model allows adequate adipocyte visualization and quantification without TAM application, but in about one third of adipocytes, it does not induce fluorescence [44]. Furthermore, adipocyte lineage tracing through temporally restricted fluorescence induction is impossible without TAM application, which is a disadvantage of this model in relation to potential in vivo experiments. On the other hand, several studies have reported adverse effects of TAM on white adipose tissue, such as browning and initial fat mass reduction followed by increased de novo adipogenesis after TAM withdrawal [45,46,47,48]. Moreover, there is no effective method of TAM elimination after application, which results in prolonged recombinase activity. This in turn leads to contradictory results when compared with experiments using doxycycline for recombination induction [47,49,50]. Therefore, to gauge the effects of TAM on epididymal AT in explant culture and to decide on a MacFat genotype for the following experiments, we stimulated epididymal white adipose tissue (WAT) explants from both homo- and heterozygous MacFat mice with 1 or 10 µM TAM for 10 days.

Microscopic imaging of the AT explants showed successful fluorescence induction through TAM as well as increased occurrence of DFAT cells through TAM stimulation in both genotypes (Figure 2A,B). Notably, in addition to fibroblast-like, stellate-shaped DFAT cells, clusters of rounder, plurivacuolar tdTo^+^ cells were observable in the TAM-stimulated explants. Flow cytometry analysis confirmed the observed rise in DFAT cells (Figure 2C,D; upper rows). While significantly more DFAT cells were detected in heterozygous MacFat AT explants as early as day 3 of cultivation with 10 µM TAM (Figure 2E), it took up to 7 days for the homozygous explants to achieve the same result (Figure 2F). After 10 days, both genotypes showed a significant DFAT cell increase for both TAM concentrations. The earlier significant increase in DFAT cells in the heterozygous genotype could be explained through the lower overall number of tdTo^+^ adipocytes in the heterozygous control condition compared with the homozygous samples. Still, under the control conditions, the proportions of DFAT cells converged for both genotypes during the cultivation period and were similar on day 10 (0.5% (homozygous) vs. 0.4% (heterozygous)), which is likely to have been caused by accumulating Cre leakage in the heterozygous explants, making both conditions comparable at that time point. However, a rise of tdTo fluorescence through TAM stimulation was expected due to the higher abundance of tdTo^+^ mature adipocytes. Interestingly, the amount of DFAT cells increased about 3.3 (homozygous) or 3.6 (heterozygous) times through 1 µM TAM and 3.7 (both genotypes) times through 10 µM TAM compared with the control condition on day 10. Hence, TAM increased the abundance of DFAT cells independent of the initial amount of tdTo^+^ mature adipocytes and directly induced adipocyte dedifferentiation in the AT explants.

It is noteworthy that while the percentage of ATMs showed no significant change between conditions (Figure 2G,H), the ratio of M1 to M2 macrophages shifted in favor of the pro-inflammatory M1 phenotype through TAM stimulation, with significant results after 10 days of cultivation (Figure 2C,D; lower rows). The phenotype shift was more pronounced in the homozygous explants, with 11.2 times more M1 macrophages than M2 under 10 µM TAM, compared with 8.7 times in the heterozygous genotype (Figure 2I,J). Due to the direct effects of TAM on dedifferentiation of mature adipocytes and macrophage activation, TAM was avoided when investigating these effects ex vivo.

### 2.3. No Signs of Increased Browning in tdTo^+^ DFAT Cells After TAM Stimulation

As stated above, the visualized tdTo^+^ stroma cells under TAM stimulation (Figure 2A,B) showed greater variance in their morphology compared with the control samples. While tdTo^+^ cells in the control condition were mostly fibroblast-like DFAT cells, the TAM-stimulated samples also contained several clusters of oval, plurivacuolar tdTo^+^ stroma cells with no or only hints of cell extensions. With the morphological similarities of these cells to beige/brown adipocytes and reported subcutaneous AT browning through TAM stimulation, we wanted to rule out the possibility of WAT browning distorting the results of the DFAT cell quantification [45].

Based on the results from the previous TAM stimulation experiments, we treated homozygous AT explants with 10 µM TAM for 10 days. To investigate the existence of brown adipocytes, immune fluorescence staining for the brown adipose tissue (BAT)-specific uncoupling protein 1 (UCP1) was performed on paraffin slices as well as whole mounts from TAM-treated AT explants (Appendix A). In contrast to BAT, there was no evident UCP1 expression in the AT explants, even after TAM stimulation. Western blot analysis of uncultivated as well as TAM-stimulated AT explants did not detect UCP1, confirming the microscopy results (Appendix A).

Therefore, the observed round, plurivacuolar tdTo^+^ cells were most probably DFAT cells in an earlier stage of the dedifferentiation process, as described by others, and not brown/beige adipocytes [17,24].

### 2.4. No Effects of TAM on Adipocyte Dedifferentiation In Vivo

With the previous results indicating TAM as an inductor of adipocyte dedifferentiation in vitro, we aimed to test this effect under in vivo conditions. Therefore, male homozygous MacFat mice were intraperitoneally injected with either 1 mg TAM or 100 µL corn oil daily for 5 consecutive days. After animal sacrifice on day 6, the epididymal and subcutaneous WAT was dissected and prepared for whole-mount microscopy. Additionally, flow cytometry analysis of the epididymal fat pad was performed.

Microscopy imaging of whole mounts showed homogenous epididymal AT with no noticeable quantity of DFAT cells (Appendix A). TAM stimulation slightly increased tdTo expression but did not alter the level of observed adipocyte dedifferentiation. Subcutaneous AT whole mounts showed isolated multilocular tdTo^+^ cell clusters, which could have been be early dedifferentiating adipocytes (Appendix A). As in the epididymal samples, TAM did not affect the number of visible DFAT cells. The marginal effect of TAM on adipocyte dedifferentiation in vivo was validated through flow cytometry analysis (Appendix A). Here, the shares of DFAT cells and ATMs as well as the ratio of pro- to anti-inflammatory macrophages was not significantly affected through TAM stimulation (Appendix A–F). Interestingly, the number of viable (DAPI-negative) cells slightly increased in the TAM-stimulated animals (Appendix A), possibly indicating increased proliferation activity or de-novo adipogenesis through TAM, as described in other publications [45,47]. 

### 2.5. Macrophage Depletion Amplifies Adipocyte Dedifferentiation in Explant Culture

ATMs are a very heterogeneous population of cells with partially contrasting effects on adipose tissue, depending on their subtype as well as the surrounding metabolic setting [7,8,11]. As dedifferentiation has previously been linked with inflammatory activity in pancreatic ß-cells as well as adipocytes, we wanted to investigate the effect of the resident tissue macrophages on DFAT cell occurrence in WAT explant culture [13,14,32].

We decided to use two different approaches for depleting ATMs: first, the colony-stimulating factor 1 receptor (CSF1R) inhibitor Plexxicon 5622 (PLX); and second, dichloromethylene bisphosphonate (clodronate)-filled liposomes. While PLX primarily induces macrophage depletion through inhibition of macrophage proliferation and differentiation, clodronate liposomes deplete macrophages by apoptosis induction after phagocytosis and intracellular clodronate accumulation [51,52]. For both methods, homozygous MacFat AT explants were cultivated under either control or macrophage-depleting conditions for up to 10 days.

Treatment with both substances showed an evident reduction in macrophages compared with the control conditions during microscopy (Figure 3A,B). Interestingly, higher frequencies of DFAT cells were noticeable under both macrophage-depleting conditions.

Flow cytometry (Figure 3C,D) confirmed the microscopy results. From day 7 onward, macrophages were significantly reduced with both PLX and clodronate liposomes. On day 1 and 3, only 40 µM PLX achieved significant macrophage reduction (Figure 3E,F). The highest depletion rates were achieved for both methods after 10 days of cultivation, with approximately the same effectiveness (ATM reduction by 81.7% via PLX or 80.8% via clodronate liposomes).

Interestingly, the macrophage depletion via clodronate liposomes coincided with a reduction in the M1–M2 ratio, while 40 µM PLX had no effect and 4 µM PLX even increased the ratio, suggesting different susceptibilities of the two macrophage subsets (Appendix A). Most importantly, both macrophage-depleted conditions showed a significantly higher share of DFAT cells in the explant culture on day 10 (Figure 2G,H). In addition, clodronate liposomes achieved higher overall proportions of DFAT cells (4.3%) compared with 40 µM PLX (2.1%), indicating higher efficiency in triggering adipocyte dedifferentiation.

While macrophage depletion decreased the overall number of SVF cells and thereby inevitably increased the share of DFAT cells, we confirmed the validity of the observed increase in DFAT cells through corrective calculations. Here, the measured DFAT cell shares were higher overall than the calculated increase in DFAT cells after macrophage depletion.

### 2.6. Bulk RNA Sequencing Data of Macrophage-Depleted AT Explants Shows Downregulation of the TNFα Signaling Pathway

With macrophage depletion increasing the number of DFAT cells in explant culture, a regulatory role of ATMs in the process of adipocyte dedifferentiation seemed plausible. To gain a better understanding of the underlying mechanisms in this process, we performed bulk RNA sequencing on AT explants cultivated with and without clodronate liposomes. Clodronate liposomes were chosen because of the overall higher levels of DFAT cells in these samples compared with PLX-depleted explants. The main goal was to discern possible regulating pathways through which macrophages may affect adipocyte dedifferentiation.

Bulk RNA sequencing revealed over 8000 significantly regulated genes (*p*-adj. < 0.05) in macrophage-depleted samples, including a plethora of signaling pathways. To narrow down these results and account for multiple testing, the FDR for the adjusted *p*-value was set at < 0.01, leaving 2836 significantly regulated genes for further examination. DAVID functional annotation analysis was then used to deduce significantly regulated KEGG pathways from the genomics data (Figure 4A), showing significant enrichment of pathways pertaining to lipid metabolism and adipocytokine signaling as well as upregulation of mature adipocyte genes (*Cebpa, Lpl, Leptin, Slc2a4*) (Figure 4B) [53,54]. Signaling pathways regulating the pluripotency of stem cells were not significantly enriched (Appendix A).

Interestingly, gene pathways associated with macrophage activity, like TNFα, NF-κ B, and chemokine signaling, were significantly enriched, as were genes associated with cytokine–cytokine receptor interactions, albeit at a lower rate than the aforementioned pathways. Closer analysis of the regulated genes downstream of the TNFα signaling pathway showed overall downregulation. The affected genes were likewise related to macrophage activity by coding for functions like pro-inflammatory cytokine (*Tnf, Il6, Il15)* and chemokine (*Ccl2, Ccl5, Cxcl1, Cxcl2, Cxcl5, Cx3cl1*) production (*Ptgs2*) as well as ECM remodeling (*Mmp3, Mmp9, Mmp14*) (Figure 4C). Genes responsible for intracellular signaling modulating immune responses, apoptosis, or cell cycle progression (*Bcl3, Nfkbia, Socs3, Traf1, Ifi47*) were also downregulated. The TGF-β and Wnt signaling pathways were not significantly enriched (Appendix A).

### 2.7. TNFα Affects Adipocyte Dedifferentiation in Explant Culture

With downregulation of the TNFα pathway coinciding with an increase in DFAT cells in explant culture after macrophage depletion, a regulatory role of ATMs through TNFα in relation to adipocyte dedifferentiation seemed probable. To verify this hypothesis, we cultivated AT explants from homozygous MacFat mice in explant culture for 10 days and stimulated them with 10 or 50 ng/mL TNFα on the third day of cultivation. PBS was used as the solvent control. Microscopic imaging on day 10 showed a reduction in DFAT cells under cytokine stimulation (Figure 5A), which was validated through the flow cytometry analysis (Figure 5B). Under TNFα stimulation, the proportion of DFAT cells significantly decreased in a concentration-dependent manner after 10 days of cultivation, with 10 ng/mL leaving two thirds and 50 ng/mL leaving one third of the share of DFAT cells in the control sample (Figure 5C). Additionally, from day 7 onward, TNFα decreased the number of ATMs in the explant culture (Figure 5D). The macrophage polarization, while predominantly statistically insignificant, showed an ambiguous pattern. The lower TNFα concentration increased the M1–M2 ratio in favor of the pro-inflammatory macrophage phenotype, whereas the high concentration decreased the ratio compared with the control, indicating the prevalence of an anti-inflammatory macrophage phenotype (Figure 5E).

With these results indicating TNFα as an inhibitor of adipocyte dedifferentiation, we wanted to verify our findings by depleting TNFα through a neutralizing antibody as a counter-control. Analogous to the stimulation experiments, AT explants from homozygous MacFat mice were cultivated for 10 days, with 1 or 10 µg/mL TNFα antibody added on day 3. Normal goat IgG was used as the isotype control. Contrary to the previous stimulation results, the flow cytometry analysis (Appendix A) showed no significant effect of the TNFα depletion on the number of macrophages in the explant culture (Appendix A). The results for the M1–M2 macrophage ratio were indistinct and again dependent on the antibody concentration. While 1 µg/mL TNFα antibody significantly decreased the M1-to-M2 quotient (Appendix A), 10 µg/mL TNFα antibody showed a slight, statistically insignificant increase in this ratio (Appendix A). Using a TNFα-neutralizing antibody did not significantly affect the level of adipocyte dedifferentiation in explant culture, regardless of antibody concentration (Appendix A). Even though the traditional threshold for statistical significance could not be reached, a trend towards increased numbers of DFAT cells was recognized after 10 days of cultivation with 10 µg/mL TNFα antibody (Appendix A).

## 3. Discussion

It is well known that metabolic stress due to obesity induces a chronic low-grade inflammatory state, which results in the dysfunction of AT mediated by proliferation and polarization of pro-inflammatory M1 macrophages [2,5,6]. In pancreatic islets, chronic inflammation leads to the loss of β-cell function by dedifferentiation into immature progenitor cells [13,14]. Contrary to long-held assumptions, mature adipocytes likewise possess the ability to dedifferentiate into fibroblast-like progenitors expressing stem cell genes and possessing pluripotent re-differentiation potential [15]. Continuous metabolic stress and activation of immunological pathways, both mediated through macrophage activity in AT, are recognized as key drivers of adipocyte dedifferentiation [7,32]. However, studies analyzing the regulation of adipocyte dedifferentiation in the context of the surrounding physiological AT stroma and microenvironment are limited.

Therefore, the main goal of our study was the identification of a regulatory mechanism through which ATMs affect adipocyte dedifferentiation in AT. Using our organotypic AT explant culture model, with the capability to preserve physiological WAT integrity and allow interaction between various cell types, and AT explants from double-reporter mice expressing GFP in ATMs and tdTo in mature adipocytes, we observed a significant increase of tdTo^+^ stellate, fibroblast-like cells resembling DFAT cells, during a 10-day cultivation period [43]. In addition, a shift of the prevalent macrophage phenotype in the explant culture from the anti-inflammatory M2 to the pro-inflammatory M1 phenotype was observed during cultivation. The preliminary assessment of both the homo- and heterozygous MacFat mouse genotypes showed increased levels of DFAT cells after TAM stimulation. Immunohistochemical staining of whole-mount and paraffin slides from stimulated AT explants as well as Western blot analysis showed no UCP1 expression in the observed fibroblast-like cells, ruling out the possibility of WAT browning. Importantly, ATM depletion through PLX and clodronate liposomes resulted in significantly increased adipocyte dedifferentiation, indicating a role of ATMs in adipocyte homeostasis. Follow-up bulk RNA sequencing of macrophage-depleted AT explants revealed significant enrichment of pathways pertaining to mature adipocyte function and macrophage signaling, with an overall downregulation of effector genes relating to TNFα signaling. While TNFα expression is not cell-specific, macrophages are a primary source of TNFα secretion and therefore, the probable cause of downregulation of the TNFα signaling pathway after ATM depletion [6,55]. In line with our hypothesis of ATMs affecting adipocyte dedifferentiation through TNFα signaling, direct stimulation of the AT explant culture with TNFα resulted in a significantly decreased share of DFAT cells. However, TNFα depletion did not significantly alter levels of DFAT cells and only slightly increased the proportion of DFAT cells under the higher antibody concentration. While the TNFα stimulation and depletion experiments were inconclusive, our findings strongly suggest a regulatory role of resident ATMs in the process of adipocyte dedifferentiation.

The adipocytic origin of the observed fibroblast-like cells was proven by the genetic composition of the murine AT explants. As stated previously, tdTo expression in the AT explants was based on an adiponectin-controlled Cre/loxP system, suggesting either independent adiponectin expression or a descent from mature adipocytes. Even though adiponectin expression is occasionally observed in other cell types, Sassmann et al. showed that the Cre-mediated recombination in AdipoqCreER^T2^ mice is highly specific for white adipose tissue [56] and consequently, so too is the fluorescence induction. Hence, the observed stellate-shaped tdTo^+^ cells were very probably of mature-adipocyte descent and, therefore, DFAT cells. In the culture model, the coinciding shift in the prevalent macrophage phenotype from the anti-inflammatory M2 to the pro-inflammatory M1 during the cultivation period was based on inherent cell death. While the method allows high-resolution visualization of AT cells and macrophages, the lack of perfusion in addition to the peripheral injuries sustained through tissue preparation result in cell death and apoptosis, which in turn activate macrophages and promote the formation of CLS. Still, the number of viable (DAPI-negative) SVF cells did not significantly decrease during the 10-day cultivation period, indicating no excessive cell death. Additionally, the number of ATMs also remained stable during cultivation. Furthermore, the AT explant culture allowed observation of adipocyte–macrophage interaction as well as direct external stimulation, making it suitable for our main research objective.

A possible explanation for increased adipocyte dedifferentiation after TAM stimulation could be reduced expression of the adipocyte differentiating factor peroxisome proliferator-activated receptor gamma (PPARγ), which is important for maintenance of the mature adipocyte phenotype [46,57,58]. Another reason for higher shares of DFAT cells might be an () increasing cell ratio due to cell death, reflecting the often-described cytotoxic effect of TAM [47]. Fittingly, we observed significantly increased polarization of macrophages into the pro-inflammatory M1 macrophage phenotype after TAM application, which may have been related to increased production of reactive oxygen species (ROS) and apoptosis induction through TAM [46].

Considering these results in addition to other known effects imposed by TAM on WAT metabolism, we decided to eliminate this possible confounder of TAM stimulation [45]. Therefore, we decided to use the homozygous MacFat genotype for all subsequent experiments regarding adipocyte dedifferentiation. Cre leakage allowed visualization of over 60% of adipocytes without prior induction in this model [44]. The problem of variance in expression between different ages and sexes was circumvented by using only male mice in a set age range. While the usage of similar lineage tracing models such as the AdipoChaser mouse, where doxycycline is used for inducing continuous lacZ expression, would allow the visualization of almost all adiponectin-expressing cells, our group has the most experience in working with MacFat mouse lines in AT explant culture [59]. Additionally, the MacFat mouse model enables visualization of macrophages without prior fluorescence induction, making it more suitable for our study purposes.

On a side note, we were not able to replicate increased levels of adipocyte dedifferentiation through TAM injection in vivo, indicating counter-regulatory mechanisms in vivo, e.g., via nerval innervation or blood flow. In line with this, Liao et al. proposed adipocyte dedifferentiation as a repair mechanism after tissue strain in AT, which could explain our findings of increased levels of DFAT cells in vitro but not in vivo [17]. In vitro, procuring AT explants inevitably coincides with tissue damage and cell death due to dissection of the fat pad, which leads in turn to decreased M2 macrophage polarization during cultivation, increased DFAT cells as a result of disturbed AT homeostasis, and an opportunity for tissue restoration. As tissue structures remain intact in vivo, adipocyte dedifferentiation is not induced to the same degree. Here, no difference in DFAT cell shares or macrophage polarization was observed, with TAM paradoxically increasing the number of viable (DAPI-negative) SVF cells. These contrary findings validated our decision to forgo TAM usage in the subsequent dedifferentiation experiments.

Both the M1 and M2 macrophage phenotypes have been associated with inhibiting adipogenesis, with Nawaz et al. describing increased levels of CD45^−^/CD31^−^/Sca-1^+^/PDGFRα^+^ adipocyte progenitors (APs) after M2 macrophage depletion and suggesting a regulatory role for ATMs keeping APs in a dormant state and preventing over-proliferation [60,61,62,63]. This aligns with our findings of increased adipocyte-derived fibroblast-like cells after macrophage depletion. While DFAT cells and adipose-derived as well as bone-marrow-derived stem cells are distinct cell populations, they express similar antigen signatures, DNA methylation, and gene expression patterns, making their distinction challenging without implementing lineage tracing [15,22,64,65].

While DFAT cells typically express stem cell markers and upregulate genes associated with cell movement, proliferation, and shape alteration, we did not observe enrichment of KEGG pathways regulating the pluripotency of stem cells in macrophage-depleted AT explants, including genes such as *Sox2*, *Nanog*, *Oct4*, c-*Myc*, and *Klf4* [26,65,66]. Additionally, we observed fold enrichment of pathways associated with mature adipocyte functions and significant upregulation of mature adipocyte genes, which are typically downregulated in DFAT cells [16,24,25]. These discrepancies between our findings and previous reports are likely to have arisen from our chosen approach being bulk RNA sequencing to analyze the collective gene expression patterns in macrophage-depleted AT, compared with the gene expression analysis of isolated DFAT cells derived from mono-layered ceiling cultures, such as has been performed in other studies. Furthermore, the upregulation of mature adipocyte genes could indicate adipogenesis in macrophage-depleted samples, possibly through re-differentiating DFAT cells [67].

As stated before, TGF-β, Wnt/β-catenin, and immunological signaling have been identified as regulatory mechanisms in adipocyte dedifferentiation [32,33,34,35,36,37,38,39]. Our results showed no significant fold enrichment of the TGF-β or Wnt signaling pathways in AT explants after macrophage depletion, ruling them out as potential regulatory mechanisms affecting adipocyte dedifferentiation in ATMs. In contrast to previous findings that identified TNFα as an inductor of adipocyte dedifferentiation, we found that downregulation of the TNFα signaling pathway after macrophage depletion coincided with an increased share of DFAT cells, while direct stimulation of AT explants with TNFα resulted in decreased DFAT cells [36,37,68]. When supraphysiological TNFα concentrations were used, we did not observe significantly decreased levels of viable (DAPI-negative) cells. A higher TNFα concentration significantly reduced the number of ATMs and it could be argued that the resulting DFAT cell increase might have been mediated not through TNFα but rather, the induced macrophage death. While macrophage death almost certainly played a role in the increasing adipocyte dedifferentiation, the lower TNFα concentration still increased DFAT cells while not significantly affecting ATMs. Nonetheless, TNFα depletion did not significantly alter ATM or DFAT cell levels; thus, the role of TNFα in adipocyte dedifferentiation in AT explant culture remains ambiguous.

Additionally, we could not identify a clear correlation between macrophage phenotype and adipocyte dedifferentiation. The increase in DFAT cell numbers after TAM stimulation coincided with an increased M1–M2 ratio, while macrophage depletion and TNFα stimulation did not significantly alter this ratio in most cases. Exceptions included the lower PLX concentration, which probably increased this ratio by activating pro-inflammatory signaling in the remaining ATMs after the death of the surrounding macrophages, and the lower TNFα concentration, as TNFα is a known inducer of the M1 macrophage phenotype [9,10]. A possible explanation for the unchanged ratio of pro- to anti-inflammatory macrophages under the higher TNFα concentration might be TNFα stronger affecting and depleting M1 macrophages, thereby reducing the M1–M2 ratio compared with the lower TNFα concentration. Accordingly, only the lower concentration of TNFα antibody significantly decreased the M1–M2 ratio, suggesting that strong TNFα depletion might activate alternative routes of M1 polarization. Furthermore, in this study, some inconsistencies might have arisen from the imprecision of the M1–M2 macrophage dichotomy, as we only used CD11c and CD301 for M1/M2 discrimination, failing to adequately reflect the multitude of macrophage subtypes. Another explanation might be that the process of adipocyte dedifferentiation in turn affects ATMs and macrophage polarization. Before DFAT cells re-enter the cell cycle and gain pluripotency, the central lipid vacuole is expelled through a process called liposecretion [27,69]. In turn, these free lipids may metabolically activate ATMs, promote pro-inflammatory polarization, and lead to the formation of CLS [70,71,72]. Then again, Lin et al. found that DFAT cells also possess immunosuppressive functions similar to stem cells, which could affect macrophage activity [73].

## 4. Materials and Methods

### 4.1. Animals

Animal experiments were performed in accordance with the rules of animal care issued by the local state authorities of Saxony, as well as German national test animal protection laws. Mice were kept in local animal facilities of Leipzig University under temperature-controlled (22 ± 2 °C), pathogen-free conditions and a 12 h light/dark cycle with free access to water and food (standard chow, Sniff GmbH, Soest, Germany).

For visualization of adipocytes and macrophages, AdipoqCreER^T2^:Rosa26-tdTo^flox/flox^ mice crossed with Csf1r-eGFP reporter mice were used [56]. This mouse strain (called MacFat mice) undergoes tamoxifen-mediated induction of red fluorescence (tdTomato) in cells expressing adiponectin (i.e., mature adipocytes) by Cre-mediated recombination, while continuously expressing the green fluorescent protein (GFP) in macrophages. However, due to tamoxifen-independent leakage of Cre, especially in the homozygous genotype and older individuals, continuous tdTomato (tdTo) expression is possible and so too, therefore, is adipocyte visualization without prior induction, as reported by us earlier [44]. Thus, mice were chosen for the respective experiments based on genotype (homo- or heterozygous), age (at least 20 weeks), and sex (male). The numbers of mice analyzed (n) and the numbers of independent experiments (N) are provided in the respective figure legends.

### 4.2. Adipose Tissue Explant Culture

Cultivation and visualization was performed entirely within the established organotypic adipose tissue culture model, called adipose tissue (AT) explants [43]. Unless otherwise stated, at least 3 independent experiments were performed with a minimum of 2 mice per trial. After animal sacrifice, the rostral epididymal white adipose tissue (WAT) was extracted under sterile conditions and subsequently cut into small (<1 mm^2^) pieces (AT explants) in phosphate-buffered saline (PBS, Biowest, Nuaillé, France). Explants were then transferred onto six-well plates with five explants per well and cultured in 1 mL to 1.5 mL RPMI1640 cell culture medium (Sigma-Aldrich, Merk KGaA, Darmstadt, Germany) supplemented with 10% fetal bovine serum (FBS), 1% insulin–transferrin–selenium (ITS) supplement (100×, Gibco, Thermo Fisher Scientific, Waltham, MA, USA), and 1% penicillin–streptomycin (P/S) mixture (10,000 U/mL, #15140-122, Gibco). Each day, for each of the experimental conditions, 2 wells (meaning 10 explants) were prepared and analyzed. Immobilization and submersion of AT explants in culture medium was ensured through fixation under sterile cell culture inserts (Millicell inserts, Merck KGaA, Darmstadt Germany) or TC inserts (Sarstedt AG&Co. KG, Nümbrecht, Germany). AT explants were cultivated over a maximal timespan of 10 days at 37 °C with 5% CO_2_ and 21% O_2_.

Microscopy imaging was performed at several timepoints during cultivation, using an FV1000 confocal microscope (Olympus, Hamburg, Germany).

### 4.3. Tamoxifen Stimulation

The selective estrogen receptor modulator tamoxifen was used to induce a time- and site-specific genetic knock-out in mature adipocytes of the AdipoqCre-ER^T2^ mouse model [56]. In explant culture, 1 µM or 10 µM tamoxifen (1 mM stock solved in ethanol, Sigma-Aldrich) was added to the culture medium on day 0. Ethanol was used as solvent control. Cultivation as described above was then performed.

For in vivo application, 50 mg tamoxifen was dissolved in 500 µL ethanol and subsequently combined with 4.5 mL corn oil to obtain a 10 mg/mL tamoxifen stock solution with 10% ethanol. Over the course of 5 days, 100 µL tamoxifen stock solution was injected intraperitoneally on a daily basis (5 mg tamoxifen in total) into each mouse. Corn oil mixed with 10% ethanol was used as solvent control. Mice were sacrificed on day 6 and epididymal and subcutaneous fat was extracted.

### 4.4. Macrophage Depletion

#### 4.4.1. Plexxicon 5622

The inhibition of the colony-stimulating factor 1 receptor (CSF1R) through the small-molecule tyrosine kinase inhibitor Plexxicon 5622 is known to be an effective way of depleting microglia cells as well as a plethora of resident tissue macrophages, including ATMs, in vivo [51,74]. Plexxicon 5622 (Plexxicon Inc., Berkeley, CA, USA) was dissolved in dimethyl sulfoxide (DMSO, Pierce^TM^, Thermo Fisher Scientific) to create a 40 mM stock solution and added to the AT explant culture medium at two timepoints (day 0 and 3) to obtain final concentrations of 4 µM and 40 µM, respectively. DMSO was used as solvent control.

#### 4.4.2. Clodronate Liposomes

Liposome-coated dichloromethylene bisphosphonate (clodronate) is a potent inhibitor of phagocytic cells like macrophages, via induction of apoptosis after ingestion and intracellular accumulation [52,75]. For application in AT explant culture, 500 µg/mL clodronate- or PBS-filled (control) lipid vehicles (both Liposoma BV, Amsterdam, The Netherlands) were first combined with RPMI1640 cell culture medium supplemented with 10% FBS, 1% ITS, and 1% P/S in 2 mL reaction tubes [76]. At day 0, AT explants were added to the clodronate- or control liposome-mixed media, respectively, and incubated for 1 h at 37 °C. For better distribution of liposomes, tubes were rotated for the duration of incubation. Afterwards, the contents of the tubes were transferred on six-well plates and cultivated as described above.

### 4.5. Tumor Necrosis Factor α (TNFα) Stimulation and Neutralization

Explants were stimulated on day 3 by adding 10 or 50 ng/mL murine recombinant TNFα (PeproTech, Thermo Fisher Scientific) dissolved in sterile H_2_O + 0.1% bovine serum albumin (BSA, lyophilized powder, #A9418, Sigma-Aldrich) to the culture media. Pure H_2_O + 0.1% BSA was used as solvent control. For cytokine neutralization experiments, 1 or 10 ng/mL TNFα antibody (AF-410-NA, R&D Systems, biotechne, Minneapolis, MN, USA) dissolved in PBS was added to the explant culture media on day 3. Normal goat IgG (AB-108-C, R&D Systems) dissolved in PBS functioned as the isotype control and was applied in equal concentrations.

### 4.6. Flow Cytometry Analysis

AT explants, either cultivated or freshly dissected, were digested using type II collagenase (Worthington Biochemical, Lakewood, NJ, USA) and incubated for 20 min at 1400 rpm and 37 °C in the thermoshaker. Digestion was stopped by putting samples on ice and adding 150 µL FCS. After discarding the lipid layer on top, the cell suspensions were filtered through a 70 µm mesh and subsequently washed twice by adding staining buffer (PBS + 3% BSA) and centrifugation (10 min, 1200 rpm, 4 °C). The remaining cell pellets were resuspended in staining buffer. Fc receptors were blocked via anti-CD16/32 (1:100, #14-0161-82, eBioscience, Thermo Fisher Scientific) and incubation on ice for 10 min. After washing and centrifugation, anti-CD301-Alexa Fluor 647 (1:50, #MCA2392, BioRad, Hercules, CA, USA) and anti-CD11c-Brilliant Violet 605 (1:100, #117334, BioLegend, BioRad) were added for 20 min on ice, for distinction of macrophage phenotypes. For cell viability assessment, 4′,6-diamidino-2-phenylindole (DAPI, 1:100, Thermo Fisher Scientific) was added 10 min prior to flow cytometry analysis. Isotype controls were prepared and measured for all experiments and used as negative controls for gating. Sample data were collected and analyzed using an BD LSRFortessa cell analyzer equipped with BD FACSDiva software 9.0 (both BD Biosciences, Franklin Lakes, NJ, USA). FlowJo software 10.10.0 (Tree Star, Ashland, OR, USA) was used for data evaluation and quantification. Flow cytometry data were gated for single cells and granulation. Subsequently, only DAPI-negative cells were further examined and divided into adipocytes/DFAT cells (GFP^−^; tdTo^+^) and ATMs (GFP^+^; tdTo^−^). The latter were then distinguished into M1 (CD11c^+^; CD301^−^) and M2 (CD11c^−^; CD301^+^) macrophages (gating strategy in Appendix A).

### 4.7. Whole Mounts and Immunofluorescence Staining

Whole mounts of epididymal and subcutaneous AT explants were prepared. Explants were washed in PBS after 10 days of cultivation and fixated in zinc formalin (21516-3.75, Polysciences Inc., Warrington, PA, USA) for 20 min. For immediate staining without prior cultivation, mice were dissected after sacrifice and the intact fat pads were fixated for 1 h in zinc formalin. Afterwards, the adipose tissue was washed in PBS and whole fat pads were cut into small pieces (<1 mm^3^). AT explants of the in vivo tamoxifen injection experiments were directly stained with DAPI (1:10,000 for 15 min) and mounted for microscopy. For uncoupling protein 1 (UCP1) staining, unspecific binding sites were blocked with staining buffer (3% normal goat serum in PBS with 1.5% BSA) for 1h and then incubated at 4 °C overnight with the primary UCP1 antibody (1:100, ab10983, Abcam, Cambridge, UK). After washing, the secondary antibody (goat-anti-rabbit Alexa Fluor 647, 1:500, #A-21245, Invitrogen, Waltham, MA, USA) was added and the mixture was incubated overnight in the dark at 4 °C. DAPI (1:10,000 for 15 min) was then used for staining the nuclei. For mounting, AT explants were washed three times with PBS and arranged on microscope slides with concave cavities. Fluorescence mounting medium (Dako, Hamburg, Germany) and cover slips were placed onto the tissue-filled cavities. Microscopy was performed using an FV1000 confocal microscope (Olympus).

### 4.8. Paraffin Embedding and Immunofluorescence Staining of Paraffin Slices

After sacrifice, the epididymal WAT fat pads of the mice were collected and fixed in zinc formalin as described above. Fixated tissue was embedded in paraffin, cut, and mounted on microscopy slides, followed by deparaffinization and unmasking in a 2100 antigen retriever (Apton Biologics Ltd., Hampshire, UK) and submersion in sodium citrate buffer for 20 min. Slides were then washed and permeabilized in PBS with 0.03% Triton-X100. Unspecific binding sites were blocked by 1:20 normal goat serum in PBS/Triton-X100 (0.03%) for 30 min. Subsequently, the primary UCP1 antibody (1:100, ab10983, Abcam, Cambridge, UK) was added, followed by incubation overnight at 4 °C. PBS with 1% BSA was used as solvent control. Secondary antibodies conjugated with either Cy3 (1:400, JIM-111-165-003, Jackson ImmunoResearch Inc., West Grove, PA, USA) or Alexa Fluor 647 (1:500, A-21245, Invitrogen) were added after washing and these samples were also incubated overnight at 4 °C in the dark. Nuclear counterstaining was performed through DAPI (1:10,000 for 5 min). Autofluorescence was extinguished using True Black^®^ lipofuscin (20× in DMF, #23007, Biotium Inc., Fremont, CA, USA) in 70% ethanol. Cover slips were mounted onto the tissue slices using fluorescence mounting medium (Dako, Hamburg, Germany). Microscopy was performed using an FV1000 confocal microscope (Olympus).

### 4.9. Western Blot Analysis

For total protein extraction from AT explants and brown adipose tissue (BAT), RIPA extraction buffer (Thermo Fisher Scientific) supplemented with phenylmethylsulfonyl fluoride (PMSF, 1%) and protease inhibitor cocktail (1:100, Roche, Basel, Switzerland) was used. After homogenization and centrifugation, samples were collected and protein concentrations assessed using a Pierce^TM^ BCA protein assessment kit (Thermo Fisher Scientific) according to the manufacturer’s instructions. Equal amounts of protein (10 µg) or blotting standard (Precision Plus Protein™ WesternC™ Blotting Standard, #1610376, BioRad) combined with sample buffer (Laemmli buffer with β-mercaptoethanol, 1:10) were loaded onto 12% SDS-PAGE gel and transferred to a nitrocellulose membrane. After blocking with 5% skim milk solution for 30 min, the blots were incubated overnight with UCP1 primary antibody (1:1000 in 5% milk solution, ab10983, Abcam, Cambridge, UK). The blot membrane was washed and incubated with horseradish peroxidase (HRP)-conjugated secondary antibody (1:5000, PI-1000, Vector Laboratories, Newark, CA, USA) and Precision Protein^TM^ StrepTactin–HRP Conjugate (for unstained protein standard detection, 1:10,000, #1610380, BioRad) for 1 h. Detection of immunoreactions was performed via visualization of peroxidase activity through an ECL kit (Pierce^TM^ ECL Western Blotting Substrate, Thermo Fisher Scientific). Blots were stripped with Western blot stripping buffer (Thermo Fisher Scientific) according to the manufacturer’s instructions and the membrane was reloaded with primary antibody against cyclophilin A (CyPA) (1:1000 in 5% milk solution, ab41684, Abcam, Cambridge, UK) for loading control. The following steps were performed as described above. Densitometry analysis comparing UCP1 and CypA expression was performed using ImageJ V1.53t (Rasband, W. S., ImageJ, U. S. National Institutes of Health, Bethesda, MD, USA). 

### 4.10. Bulk RNA Sequencing

After cultivating AT explants with clodronate or PBS liposomes (500 µg/mL respectively) for 10 days, explants were snap frozen and stored at −80 °C until sequencing. RNA extraction, library preparation, and bulk RNA sequencing were performed by GENEWIZ Europe (Azenta Life Sciences, Leipzig, Germany) following the CELseq2 protocol [77]. Gene expression was analyzed using the Deseq2 algorithm on GALAXY servers (https://academic.oup.com/nar/article/50/W1/W345/6572001; accessed on 24 February 2023). DAVID Functional Annotation Bioinformatics Microarray Analysis (https://david.ncifcrf.gov; accessed on 24 February 2023) was used for identification of relevant KEGG pathways and associated genes [53,54].

### 4.11. Statistical Analysis

GraphPad Prism 10.2.0 (GraphPad Software, Inc., Boston, MA, USA) was used to perform statistical analyses. Data were tested for normality using the Shapiro–Wilk test. For normally distributed data, ROUT outlier tests (Q = 1%) were performed. Further analysis was executed in accordance with normality, either through a two-tailed Student’s *t*-test (data with normal distribution) or Wilcoxon test (data with non-normal distribution); *p*-values < 0.05 were considered statistically significant. Data are shown as means ± SEM. The sample size was estimated based on previous experiments and is stated in the respective legends. For qualitative analyses, e.g., microscopy, at least three independent experiments were performed. No additional blinding of the respective investigator was performed.

## 5. Conclusions

In summary, our findings showed a correlation between ATM activity and adipocyte dedifferentiation in AT explant culture, with ATMs inhibiting excessive adipocyte dedifferentiation. While TNFα signaling did affect DFAT cell levels, our findings were inconclusive overall and further studies of the regulating pathways in adipocyte dedifferentiation are needed.

Although the chosen explant culture model allowed cellular functions to be analyzed in the context of intact adipose tissue, it was limited by the overall induced cell death and predominantly proinflammatory setting. Therefore, analyzing the mechanisms of adipocyte dedifferentiation in a physiological context in vivo is essential for further research. Going forward, more precise classification of the involved macrophage subtypes is also required, as we used only CD11c and CD301 markers for macrophage classification, which does not reflect in depth the intricacy of this cell type.

Additionally, investigating the effects of DFAT cells on the surrounding immunological cells and tissue structures might be beneficial for improving our understanding of the physiological role of this cell type and potential application in the field of regenerative medicine.

## Figures and Tables

**Figure 1 ijms-25-13019-f001:**
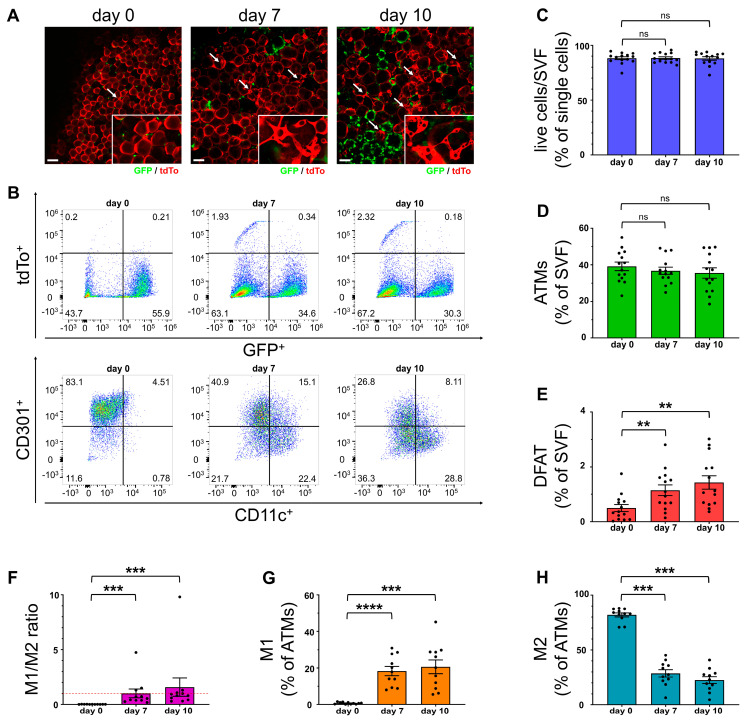
Adipocyte dedifferentiation in AT explant culture. Epididymal white adipose tissue (WAT) explants from homozygous MacFat mice (n = 11–14, N = 5–6) were cultivated under standard cultivation conditions for 10 days. (**A**) Microscopy analysis of adipose tissue (AT) explants from MacFat mice (red: adipocytes; green: adipose tissue macrophages (ATMs); arrow: DFAT cell) in explant culture at different timepoints. (**B**) Representative flow cytometry dot plots showing increase in dedifferentiated fat (DFAT) cells (GFP^−^; tdTo^+^) (upper row) and a shift from anti- (CD11c^−^; CD301^+^) to pro-inflammatory (CD11c^+^; CD301^−^) ATM phenotype (lower row). (**C**) Proportion of live cells (4′,6-diamidino-2-phenylindole (DAPI)-negative, n = 14, N = 6) from single cells. (**D**,**E**) Proportion of (**D**) ATMs (GFP^+^; tdTo^−^, n = 14, N = 6) and (**E**) DFAT cells (GFP^−^; tdTo^+^, n = 14, N = 6) from the stromal vascular fraction (SVF). (**F**) Ratio of pro- (M1) to anti-inflammatory (M2) macrophages (n = 11, N = 5). Red line marks a balanced 1:1 ratio. (**G**,**H**) Proportion of (**G**) pro- (M1, CD11c^+^; CD301^−^, n = 11, N = 5) and (**H**) anti-inflammatory (M2, CD11c^−^; CD301^+^, n = 11, N = 5) macrophages from ATMs. Data shown as means ± SEM. Scale bars = 100 µm. ** *p*-value < 0.01; *** *p*-value < 0.001; **** *p*-value < 0.0001, ns means not significantly.

**Figure 2 ijms-25-13019-f002:**
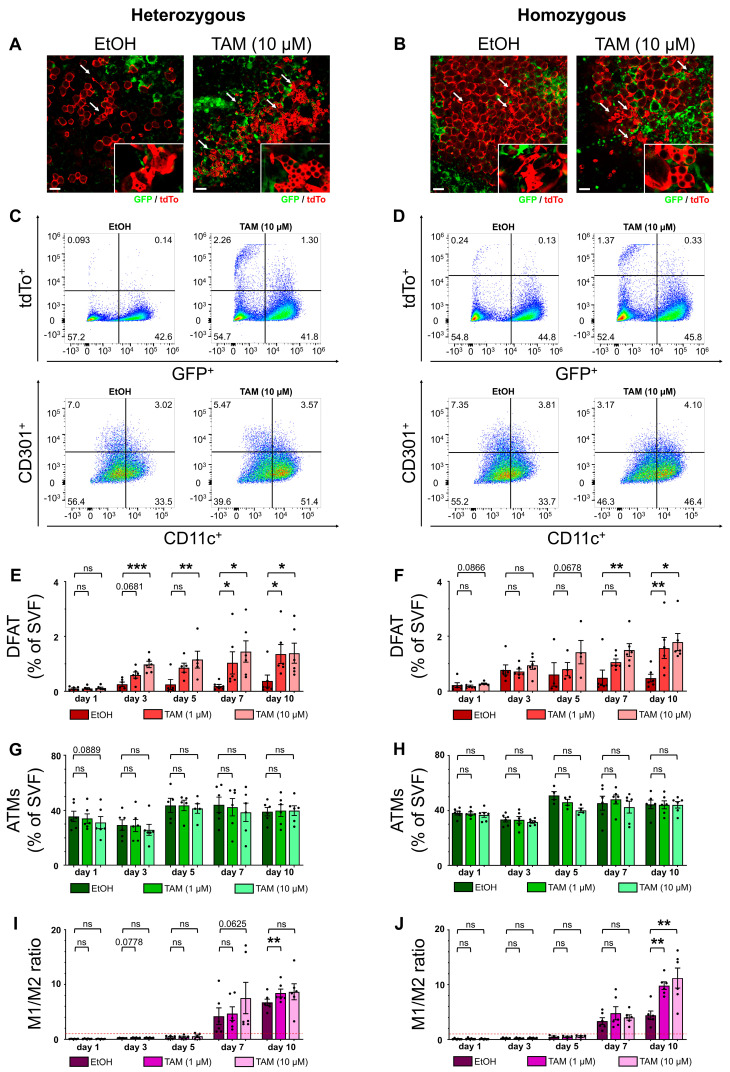
Effects of tamoxifen (TAM) on adipocyte dedifferentiation and ATM phenotype, comparing homo- and heterozygous MacFat mice. Epididymal WAT explants from homo- and heterozygous MacFat mice (n = 4–6, N = 3 per genotype) were stimulated with 1 or 10 µM TAM or 10 µL ethanol (EtOH, control) for 10 days. (**A**,**B**) Microscopic imaging of TAM-stimulated (10 µM TAM) and non-stimulated (EtOH) AT explants from (**A**) hetero- and (**B**) homozygous MacFat mice (red: adipocytes; green: ATMs; arrow_ DFAT cells) on day 10 in explant culture. (**C**,**D**) Representative flow cytometry dot plots of (**C**) hetero- and (**D**) homozygous AT explants showing DFAT cell (GFP^−^; tdTo^+^) increase (upper rows) and a shift from anti- (CD11c^−^; CD301^+^) to pro-inflammatory (CD11c^+^; CD301^−^) ATM phenotype (lower rows) under TAM stimulation on day 10 in explant culture. (**E**,**F**) Proportion of DFAT cells (GFP^−^; tdTo^+^) from SVF at different timepoints during cultivation of (**E**) hetero- (n = 5–6) and (**F**) homozygous (n = 4–6) AT explants under TAM stimulation. (**G**,**H**) Proportion of ATMs (GFP^+^; tdTo^−^) from SVF on different timepoints during cultivation of (**G**) hetero- (n = 5–6) and (**H**) homozygous (n = 4–6) AT explants under TAM stimulation. (**I**,**J**) Ratio of pro- (M1) to anti-inflammatory (M2) macrophages at different timepoints during cultivation of (**I**) hetero- (n = 5–6) and (**J**) homozygous (n = 4–6) AT explants under TAM stimulation. Red line marks a balanced 1:1 ratio. Data represented as means ± SEM. Scale bars = 100 µm. * *p*-value < 0.05; ** *p*-value < 0.01; *** *p*-value < 0.001, ns means not significantly.

**Figure 3 ijms-25-13019-f003:**
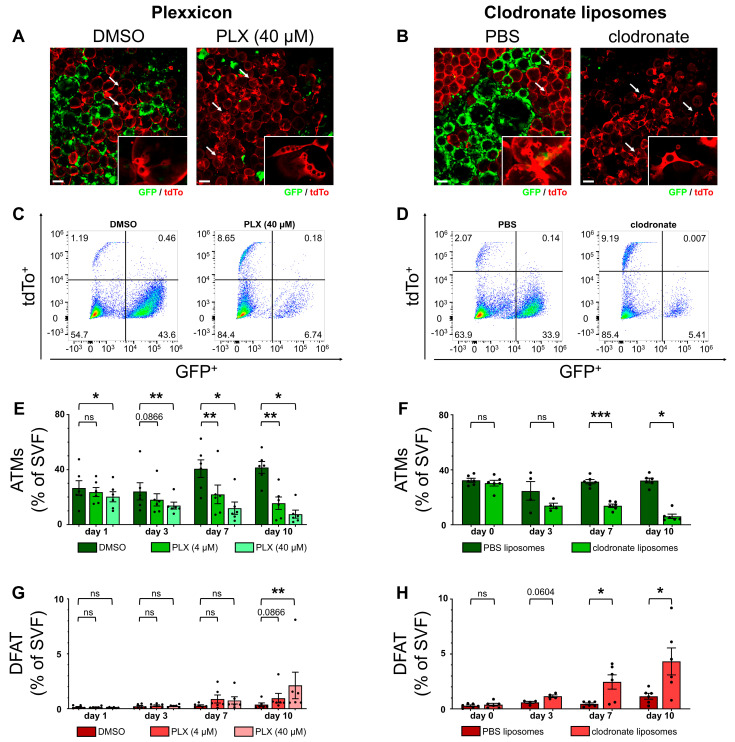
ATM depletion through Plexxicon 5622 (PLX) or clodronate liposomes amplifies DFAT cell levels. Epididymal WAT explants from homozygous MacFat mice (n = 6, N = 3 per condition) were cultivated under ATM depleting conditions with PLX or clodronate liposomes for 10 days. (**A**,**B**) Microscopic imaging on day 10 of cultivation of AT explants (red: adipocytes; green: ATMs; arrow: DFAT cells) stimulated with (**A**) dimethyl sulfoxide (DMSO, ctrl)/40 µM PLX or (**B**) PBS liposomes (ctrl)/clodronate liposomes. (**C**,**D**) Representative flow cytometry dot plots showing DFAT cell (GFP^−^; tdTo^+^) increase on day 10 after (**C**) PLX or (**D**) clodronate liposome stimulation. (**E**,**F**) Decreasing proportion of ATMs (GFP^+^; tdTo^−^) from SVF with (**E**) PLX or (**F**) clodronate liposomes. (**G**,**H**) Increasing numbers of DFAT cells (GFP^−^; tdTo^+^) coinciding with (**G**) PLX or (**H**) clodronate liposome stimulation. Data shown as means ±SEM. Scale bars = 100 µm. * *p*-value < 0.05; ** *p*-value < 0.01; *** *p*-value < 0.001; ns means not significantly.

**Figure 4 ijms-25-13019-f004:**
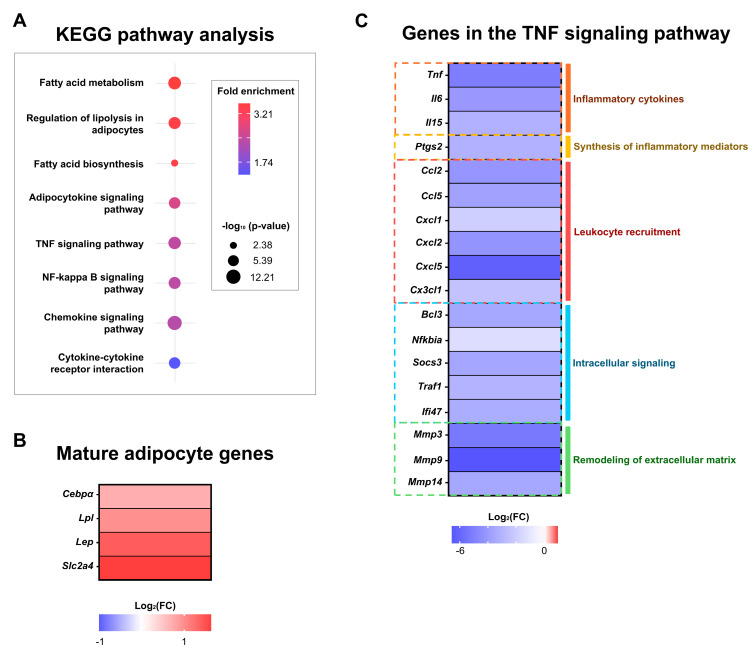
Bulk RNA sequencing of macrophage-depleted AT explants. Epididymal WAT explants from homozygous MacFat mice (n = 4, N = 2) were cultivated under ATM-depleting conditions with clodronate liposomes for 10 days and analyzed via bulk RNA sequencing. (**A**) Pathway analysis of the gene expression profile in macrophage-depleted AT explants. (**B**) Heat map of genes coding for mature adipocytes. (**C**) Heat map of genes coding for inflammatory cytokines, chemokines, and enzymes regulated through the TNFα signaling pathway.

**Figure 5 ijms-25-13019-f005:**
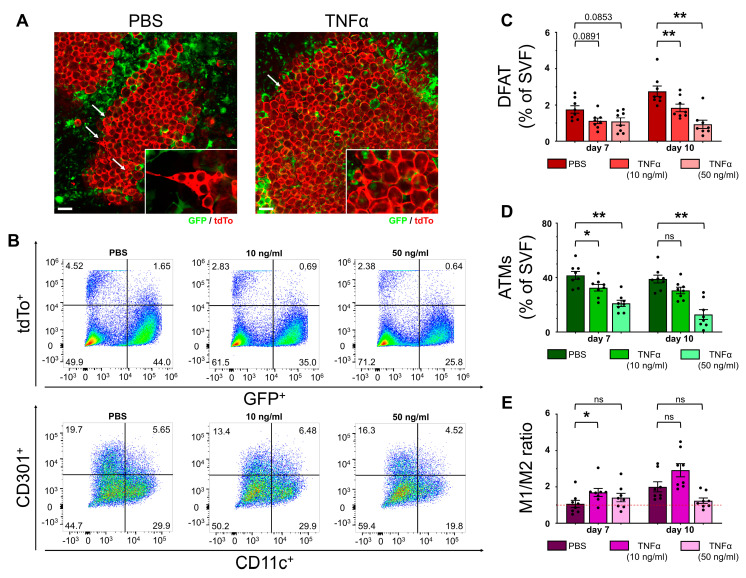
TNFα signaling affects adipocyte dedifferentiation in explant culture. Epididymal WAT explants from homozygous MacFat mice (n = 8, N = 4) were stimulated with 10 or 50 ng/mL TNFα and cultivated for 10 days. (**A**) Microscopy imaging of TNFα-stimulated and non-stimulated AT explants (red: adipocytes; green: ATMs; arrow: DFAT cells) on day 10 of cultivation. (**B**) Representative flow cytometry dot plots showing DFAT cells (GFP^−^; tdTo^+^) and ATMs (GFP^+^; tdTo^−^) (upper row) as well as the ratio of pro- (CD11c^+^; CD301^−^) to anti-inflammatory (CD11c^−^; CD301^+^) macrophages (lower row). (**C**) Proportion of DFAT cells (GFP^−^; tdTo^+^) from SVF at different timepoints during cultivation after TNFα stimulation. (**D**) Proportion of ATMs (GFP^+^; tdTo^−^) from SVF at different timepoints during cultivation after TNFα stimulation. (**E**) Ratio of pro- (M1) to anti-inflammatory (M2) macrophages at different timepoints during cultivation after TNFα stimulation. Red line indicates a balanced 1:1 ratio. Data shown as means ± SEM. Scale bars = 100 µm. * *p*-value < 0.05; ** *p*-value < 0.01, ns means not significant.

## Data Availability

The data presented in this study are available upon request from the corresponding author.

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
