# Peer review of "The Impact of Resident Adipose Tissue Macrophages on Adipocyte Homeostasis and Dedifferentiation"

_ijms, 2024, doi:10.3390/ijms252313019_

Round 1
Reviewer 1 Report
Comments and Suggestions for Authors
Manuscript ID: ijms-3312254
Resident adipose tissue macrophages impact on adipocyte ho-2 meostasis and dedifferentiation
It is an interesting work, I consider that this manuscript provides evidence that will help to understand the mechanisms involved in obesity as a pathological state and their alternative therapeutic targets, which could help to counteract this health problem worldwide in the future.
Line 537 In point 4.2 it would be important to add the "experimental design" section, where you could tell us how many groups are being analysed and the number of animals or samples per group.
Line 670-675 What was the method used to perform the relative quantification of the spots or what was the software used to perform the densitometry?
Author Response
“Line 537 In point 4.2 it would be important to add the "experimental design" section, where you could tell us how many groups are being analysed and the number of animals or samples per group.”
Thank you for bringing this to our attention. As the number of animals and trial varies slightly from experiment to experiment, we found it sensible to add the concrete number of animals (n) and experiments performed (N) into the according figure description texts (highlighted yellow in revised publication). Animals were split evenly over the performed experiments. Predominantly, 3 independent experiments with 2 mice per group (6 mice in total) were performed. Per condition and day, 2 wells (meaning 10 explants) were prepared and analyzed. We revised point 4.1 (line 541-542) and 4.2 (line 545-546 and line 554-555) to include these statements.
“Line 670-675 What was the method used to perform the relative quantification of the spots or what was the software used to perform the densitometry.”
Thank you for bringing this to our attention. As no visible UCP-1 expression was observed in white adipose tissue on Western blot, we did not initially perform a densitometry. For unambiguity of our results, we now analyzed the Western blot images through ImageJ and added the densitometric analysis to point 4.9 of our materials and methods section (line 683-685). Additionally, a complete table of densitometry results was added to the uncropped Western blot images.
Reviewer 2 Report
Comments and Suggestions for Authors
Resident adipose tissue macrophages impact on adipocyte homeostasis and dedifferentiation
Topic is very interesting. Macrophage infiltration into adipose tissue is a key pathological factor inducing adipose tissue dysfunction and contributing to obesity-induced inflammation and metabolic disorders.
Since an animal experiment is involved in this study I would like to ask if the authors
have provided an ARRIVE checklist according to Ethical Guidelines.
During the review process I have checked if the animal experiments were performed in accordance with the rules of animal care issued by the local state authorities of Saxony as well as German national test animal protection laws. Mice were kept in local animal facilities of Leipzig University under temper- ature controlled (22 ± 2°C), pathogen-free conditions at a 12-h light/dark cycle with free 525 access to water and food (standard chow, Sniff GmbH, Soest, Germany). The article has provided detailed information about the animal
experimentation process; It must be checked if there are any ethical or animal welfare concerns, and 3) whether it complies with the commonly accepted
'3R' principles: Replacement of animals with alternatives wherever
possible; Reduction in the number of animals used; Refinement of
experimental conditions and procedures to minimize the harm to animals. Please demonstrate the required information about animal studies. The working methods and statistics are chosen according to the purpose of the study.
Author Response
“Since an animal experiment is involved in this study I would like to ask if the authors
have provided an ARRIVE checklist according to Ethical Guidelines. It must be checked if there are any ethical or animal welfare concerns, and 3) whether it complies with the commonly accepted '3R' principles: Replacement of animals with alternatives wherever
possible; Reduction in the number of animals used; Refinement of
experimental conditions and procedures to minimize the harm to animals. Please demonstrate the required information about animal studies.“
We agree with your notion of the importance of animal welfare in scientific research. While the intricate metabolic processes of adipose tissue cannot be replicated and examined without animal sacrifice at this point in time, the used adipose tissue explant culture model drastically reduces the number of needed animals. As the epididymal fat pad is dissected and distributed onto several well plates, various stimulatory conditions can be analyzed using a single mouse. Additionally, the homozygous genotype of the MacFat mouse model allows for adipocyte visualization without prior tamoxifen stimulation in vivo, meaning overall less stress for the animals. Therefore, we believe that the used adipose tissue explant culture model complies with the accepted 3R principles and heeds animal welfare.
As we provide ample information pertaining to our standards of animal care and followed ethical guidelines in section 4.1 of our materials and methods (line 527-531) and institutional review board statement (line 739-741), we did not initially provide a separate ARRIVE checklist with our submission. We now composed an ARRIVE Essential 10 checklist to verify the reliability of our findings.
Reviewer 3 Report
Comments and Suggestions for Authors
The article addresses a topic of interest to readers of the journal, namely the effect of adipose tissue macrophages on adipocyte homeostasis and dedifferentiation.
The article is quite interesting and well-crafted, although I would like to make a few observations:
The introduction is appropriate and provides a solid context for the topic. However, the bibliographic references supporting it are outdated, with an obsolescence index (median age of the citations) of 9 years, which is considered high.
I believe the methodology section, where the research process is explained in detail so that any researcher can replicate the experiment by following the described steps, should be placed after the introduction rather than after the results. Additionally, inclusion criteria are missing.
The results section is extensive, very well-developed, and supported by graphs that greatly facilitate understanding.
The discussion is quite comprehensive and allows readers to compare the results with those obtained by other authors. However, a section on the strengths and limitations of the study is missing.
The conclusions are precise, concise, and very appropriate.
The general bibliography, not just that of the introduction, also shows a high obsolescence
Author Response
“The introduction is appropriate and provides a solid context for the topic. However, the bibliographic references supporting it are outdated, with an obsolescence index (median age of the citations) of 9 years, which is considered high. The general bibliography, not just that of the introduction, also shows a high obsolescence.”
Thank you for bringing this to our attention. We revised the literature and added several recent publications (highlighted in green in revised publication), to support the validity of our findings. However, new studies on the topic of mature adipocyte dedifferentiation are limited as Wang et al. (2018, PMID: 29909970) only proofed the occurrence of dedifferentiated adipocytes in physiological circumstances through lineage tracing 6 years ago. Therefore, we would like to note that the topic of our work is very recent and only starting to gain the interest of a broader audience. Therefore, the seemingly older publications are often the most recent ones, especially concerning the molecular basic research of dedifferentiated fat cells. Newer publications almost always focus the clinical application of DFAT cells, which, while important, is not the main focus of our research paper.
“I believe the methodology section, where the research process is explained in detail so that any researcher can replicate the experiment by following the described steps, should be placed after the introduction rather than after the results. Additionally, inclusion criteria are missing.”
The order of the different sections is specified by the IJMS, where it is stated that the materials and methods should come after the discussion. We understand your reasoning for wanting to change the order, but as we want to comply with the journals guidelines, we would like to keep the section positions as they are.
As stated in our results section, mice were chosen by age and genotype. As we only used male mice, we also added this to point 4.1 (line 539-541). We analyzed the flow cytometry results for outliers (ROUT, Q = 1%) and excluded only values marked as such (point 4.11, line 697-698). Otherwise, all measured results were included in our study and no other inclusion criteria were applied.
The only exception is in point 2.2, where three mice had to be excluded from analysis on day 5 due to technical error during flow cytometry analysis with incorrect and incomparable measurements (figure 2, line 186-202). As this was an isolated occurrence and not a decision based on inclusion criteria, we excluded these values without further comment. Furthermore, statistical significant changes were observable on day 7 and 10, where values from all six mice were included.
“The discussion is quite comprehensive and allows readers to compare the results with those obtained by other authors. However, a section on the strengths and limitations of the study is missing.”
Thank you for bringing this to our attention. As we already consider strength and limitations in our discussion, we did not initially add a separate strengths and limitations section. We now added a concise strengths and limitations section as part of the conclusions under point 5 (line 712-718) for readers to quickly grasp the limits of our work.
Reviewer 4 Report
Comments and Suggestions for Authors
The manuscript is focused on the analysis of the effect of adipose tissue macrophages activity on adipocyte dedifferentiation tested in adipose tissue explant culture. The authors performed a series of experiments showing that resident macrophages play a role in adipocyte homeostasis, However, results on the role of TNFα signaling in adipocyte differentiation have been inconclusive.
The study is carefully conducted, the methods are adequate, the results are well documented, and the literature data is up-to-date.
I have only minor comments:
1) Comparing pancreatic to adipocytes is somewhat puzzling. While β-cells synthesize and secrete insulin, adipocytes are particularly important for triglyceride accumulation and maintenance of energy homeostasis.
2) The authors should define a dedifferentiated adipocyte. The main task of adipocytes is to respond to stimulating levels of insulin leading to increased glucose uptake, which has not been demonstrated in fibroblasts. Do dedifferentiated adipocytes have changes in genes in key transcription factors, glucose metabolism genes and others?
3) There is a large difference in the amount of macrophages and their product TNFα in the adipose tissue of non-obese and obese animals. What situation do the authors' results indicate?
4) The conclusion of the study should be rewritten to make it more understandable.
5) Check the spelling of Plexxicon, which is sometimes written with one or two x.
Author Response
“1) Comparing pancreatic to adipocytes is somewhat puzzling. While β-cells synthesize and secrete insulin, adipocytes are particularly important for triglyceride accumulation and maintenance of energy homeostasis.”
We thank the reviewer for pointing out this misunderstanding. It was never our intention to directly compare pancreatic ß-cells and white adipocytes, since they are, as pointed out by the reviewer, very distinct cell populations. Our main goal in mentioning pancreatic ß-cells was highlighting the underlying mechanisms of dedifferentiation and subsequent loss of function through oxidative stress, inflammation and hypoxia as well as address the role of dedifferentiation in a pathophysiological context (line 48-51). These mechanisms are not specific to ß-cell dedifferentiation and pro-inflammatory stimuli have also been observed to induce adipocyte dedifferentiation (line 69-72).
“2) The authors should define a dedifferentiated adipocyte. The main task of adipocytes is to respond to stimulating levels of insulin leading to increased glucose uptake, which has not been demonstrated in fibroblasts. Do dedifferentiated adipocytes have changes in genes in key transcription factors, glucose metabolism genes and others?”
Thank you for bringing this to our attention. In our introduction, we define dedifferentiated adipocytes as fibroblast-like (line 52) cells expressing stem cell genes and surface markers (line 52-53). We also state their pluripotent redifferentiation potential (line 63-64). We added a section to describe the metabolic and secretory abilities of dedifferentiated adipocytes (line 60-63) to better illustrate their cellular functions and adequately define this cell type.
“3) There is a large difference in the amount of macrophages and their product TNFα in the adipose tissue of non-obese and obese animals. What situation do the authors' results indicate?”
Since obesity is associated with an accumulation of adipose tissue macrophages, it could be argued that adipocyte dedifferentiation would decrease as our study indicates a regulatory role of adipose tissue macrophages on the dedifferentiation process. This, however, would ignore the aspect of macrophage phenotype, which decisively affects the macrophages function and effect on its environment. As obesity is associated with a decreasing number of regulatory, anti-inflammatory M2 macrophages as well as an overall disruption the adipose tissue homeostasis with constant metabolic strain on all cell types, we think that obesity would potentially increases the level of adipocyte dedifferentiation. As the role of TNFα in adipocyte dedifferentiation remained inconclusive in our research, it is difficult for us to speculate on its effect in an obese setting.
This, however, is merely speculation as we did not use adipose tissue of obese individuals for any of our experiments or alter the metabolic state in our explant culture. Therefore, we are unsure how adding this aspect would be beneficial to the overall scientific quality of our publication.
“4) The conclusion of the study should be rewritten to make it more understandable.”
Please note that our conclusion is a separate section from the discussion and is stated behind the methods and materials (point 5, line 706-722) as the IJMS submission template indicated. Here, we clearly state our main findings of correlation between ATM activity and adipocyte dedifferentiation as well as the ambiguous role of TNFα in this process. Furthermore, we now address our studies limitations and give an outlook on further possible research subjects our study suggests.
“5) Check the spelling of Plexxicon, which is sometimes written with one or two x.”
Thank you for bringing this to our attention. We revised the paper so that Plexxicon should always be written with two x.
Round 2
Reviewer 3 Report
Comments and Suggestions for Authors
Is ok now